# Integrins: An Important Link between Angiogenesis, Inflammation and Eye Diseases

**DOI:** 10.3390/cells10071703

**Published:** 2021-07-06

**Authors:** Małgorzata Mrugacz, Anna Bryl, Mariusz Falkowski, Katarzyna Zorena

**Affiliations:** 1Department of Ophthalmology and Eye Rehabilitation, Medical University of Bialystok, 15-089 Bialystok, Poland; anna.bryl@umb.edu.pl; 2Medical University of Bialystok, 15-089 Bialystok, Poland; mariusz.falkowski@adres.pl; 3Department of Immunobiology and Environmental Microbiology, Medical University of Gdansk, 80-211 Gdansk, Poland; katarzyna.zorena@gumed.edu.pl

**Keywords:** integrins, eye, cornea, dry eye disease, cataract, glaucoma, retina, age macular degeneration, diabetic retinopathy, retinopathy of prematurity, high myopia

## Abstract

Integrins belong to a group of cell adhesion molecules (CAMs) which is a large group of membrane-bound proteins. They are responsible for cell attachment to the extracellular matrix (ECM) and signal transduction from the ECM to the cells. Integrins take part in many other biological activities, such as extravasation, cell-to-cell adhesion, migration, cytokine activation and release, and act as receptors for some viruses, including severe acute respiratory syndrome-related coronavirus 2 (SARS-CoV-2). They play a pivotal role in cell proliferation, migration, apoptosis, tissue repair and are involved in the processes that are crucial to infection, inflammation and angiogenesis. Integrins have an important part in normal development and tissue homeostasis, and also in the development of pathological processes in the eye. This review presents the available evidence from human and animal research into integrin structure, classification, function and their role in inflammation, infection and angiogenesis in ocular diseases. Integrin receptors and ligands are clinically interesting and may be promising as new therapeutic targets in the treatment of some eye disorders.

## 1. Introduction

Integrins belong to a group of cell adhesion molecules (CAMs) which is a large group of membrane-bound proteins. They are involved both in cell attachment to the extracellular matrix (ECM) and in signal transduction from the ECM to the cells. They also take part in numerous biological activities, namely extravasation, cell-to-cell adhesion, cell migration, and function as receptors for certain viruses, including adenovirus, echovirus, hantavirus, foot-and-mouth disease, polio virus and severe acute respiratory syndrome-related coronavirus 2 (SARS-CoV-2). CAMs also include selectins, cadherins, immunoglobulin superfamily and other molecules, including CD44. Cell adhesion molecules are classified using the CD nomenclature (from 1 to 130) [1,2,3]. Integrins receive and transmit biochemical and mechanical signals through the cell membrane in both directions. Signals which develop inside the cell lead to conformational changes of the molecule and transmission of integrin into a state which enables ligand binding. The name “integrins” has been coined to denote the function they have maintaining a multicellular organism as a whole. They significantly affect the integrity of the cytoskeleton–ECM connections.

Knowing the structure, function and mechanism of integrin activation is very important, allowing detection of new compounds to be used in the case of inflammation, fighting thrombotic processes. It also enables searching for new possibilities in the case of venous embolism, neoplastic metastases and other diseases.

## 2. Integrin Structure

Integrins are heterodimeric glycoproteins which serve a function of transmembrane receptors. They are composed of two chains, α and β, which can be joined together making 24 combinations of various heterodimers. These chains are noncovalently associated. The spatial structure of integrins resembles the human body [1,2,3,4]. The “head” is based on two “legs” which are changed into the transmembrane domain. Interactions between the α and β subunits usually take place in the “head.” The α chain is composed of a seven-bladed β-propeller joined with a thigh, calf-1 and calf-2 domains. This structure supports the integrin head. The last three or four blades of the β-propeller include domains that bind Ca^2+^. The β chain is composed of βI, hybrid and PSI (plexin/semaphorin/integrin) domains and four cysteine-rich epidermal growth factor-like (EGF) modules. The βI domain contains Mg^2+^. Divalent ions are essential for normal function of integrins. Each ion has a different function. Mn^2+^ and Mg^2+^ activate adhesion processes. Ca^2+^ ions, depending on concentration, may have an inhibitory or stimulatory effect. High Ca^2+^ concentration inhibits adhesion, whereas low Ca^2+^ concentration with optimal Mg^2+^ concentration stimulates binding of a ligand with an integrin [5].

## 3. Integrin Activation

Knowing the mechanism of integrin activation enables searching for new therapies of many diseases, e.g., cardiovascular diseases (venous emboli, myocardial infarctions), inflammatory diseases, allergies and metastatic processes. The presence and activity of integrins depend on numerous factors, which can be activators or inhibitors. These include hormones, cytokines, mediators of systemic inflammation, active components of the complement system, active oxygen species, endotoxins or pharmacological compounds. A change in the expression of integrin receptors usually takes place at the transcription level. Integrin activation leads to molecular transformation, which enables ligand binding [2].

Intracellular domains of both chains bind directly or indirectly with the cellular actin-based cytoskeleton [1,6]. Depending on the signal direction, integrin activation may take place in two ways: as outside–in signaling or inside–out signaling. In the case of outside–in signal transduction, ligand binding to extracellular domains of an integrin causes a shift of the subunit “head” segments, elongation of the integrin in the “knee” region and extending the whole structure. As a result, the domains forming the “legs” move apart. The transmembrane and cytoplasmic regions of both subunits are moved apart, too. This way, the integrin molecule is activated, showing a higher affinity to ligands. Then, clustering occurs, which in turn initiates adhesion processes between the cell and the extracellular matrix. In the case of inside–out activation, ligand conjugation is not necessary to activate an integrin. Such activation is mediated by intracellular proteins bound to the cytoplasmic skeleton: kindlin, talin and migflin [2].

Integrins occur in three various spatial configurations: closed, activated (extended-closed) and active (extended-open) conformations. During transition from a closed to activated conformation, there occurs extension of the integrin between calf-1 and thigh in the α subunit and between I–EGF-1 and I–EGF-2 in the β subunit [7].

The integrin molecule is rich in cysteine residues which form disulfide bridges. A transition of an integrin from its inactive form to a form enabling ligand binding is preceded by reorganization of disulfide bonds inside the molecule. This reaction is catalyzed by protein disulfide isomerase (PDI). An example of PDI participation in integrin activation is the process of platelet aggregation. The PDI count is much lower in inactive platelets than during activation. Using anti-PDI antibodies, inhibition of blood platelet aggregation was observed. Other examples involve processes taking place in vascular endothelial cells. Mn^2+^ ions initiate formation of relatively stable complexes between the α_V_β_3_ integrin and PDI [8].

## 4. Integrin Classification

Vertebrates have 18 α and eight β subunits, which form various heterodimers. Integrin heterodimers contain a number of combinations of α and β subunits. Depending on the type of the ligand bound, integrins can be classified as collagen-binding integrins (α_1_β_1_, α_2_β_1_, α_10_β_1_, α_11_β_1_), integrins recognizing the RGD motif (the triple amino acid sequence arginine–glycine–aspartic acid (α_5_β_1_, α_V_β_1_, α_V_β_3_, α_V_β_5_, α_V_β_6_, α_V_β_8_, α_IIb_β_3_, α_8_β_1_)), laminin-binding integrins (α_3_β_1_, α_6_β_1_, α_7_β_1_, α_6_β_4_) and leukocyte-binding integrins (α_L_β_2_, α_M_β_2_, α_X_β_2_, α_D_β_2_). The β_2_ integrin subunit (CD18) is able to couple to one of the α subunits (α_L_-CD11a, α_M_-CD11b, α_X_-CD11c, α_D_-CD11d) to form lymphocyte function-associated antigen-1 (LFA-1/α_L_β2/CD11a/CD18), macrophage-1 antigen/complement receptor 3 (Mac1/CR3/α_M_β2/CD11b/CD18) and complement receptor 4 (p150,95/CR4/CD11c/CD18). The CD11a/CD18 is present mainly on all leukocytes, whereas CD11b/CD18, CD11c/CD18 and CD11d/CD18 are found on myeloid cells. The α_M_β_2_ integrin (also known as CR3, CD11b/CD18 or Mac-1) is expressed on phagocytic cells and engaged in the adhesion of leukocytes to the endothelium and microbial opsonization. Ligands for CR3 contain the complement component iC3b, the intercellular adhesion molecule (ICAM-1) and coagulation factors such as fibrinogen and factor X [1,2,9,10,11].

Another classification is based on the presence of the αI domain. Belonging to one of the integrin families depends on the β chain (with the most important β_1_, β_2_ and β_7_ chains) combined with various α chains. Each of the β subunits may form a heterodimeric receptor with various α subunits. An exception is the α_V_ subunit, which binds several different β subunits, e.g., α_V_β_1_, α_V_β_3_, α_V_β_5_, α_V_β_6_. Since they may appear at 2–7 weeks after lymphocyte stimulation, β_1_ integrins are called very late antigens (VLA) or CD29. They participate in the binding of cells with the extracellular matrix. They bind laminin, fibronectin, vitronectin, collagen and other proteins of the extracellular matrix; therefore, they serve an important function in cell adhesion to the background. They occur on numerous cells of the immune system. They are absent on erythrocytes. Eosinophils show a presence of the α_4_β_1_ and α_6_β_1_ integrins, α_4_β_1_ and α_5_β_1_ basophils, α_3_β_1,_ α_4_β_1_ and α_5_β_1_ mastocytes. Neutrophils contain all the β1 integrins, except for the α_4_β_1_ integrin. On the other hand, β_2_ integrins are present on cell membranes of all leukocyte populations. This subunit can be linked with one of the three α subunits which form the CD11a, CD11b, CD11c titer. The group of β_2_ integrins contains the LFA-1, Mac-1 and GP-150/95 glycoproteins. The CD11a/CD18 integrin is called LFA-1 (leukocyte function-associated antigen) since these antigens are found only on leukocytes. The Mac-1 integrins occur in their inactive form on neutrophils, monocytes and NK cells. Activation and binding with ICAM-1 is caused by inflammatory factors. As a result, neutrophils are bound to endothelial cells [12,13].

Integrin ligands also include receptors, which belong to a family of immunoglobulin-like CAMs such as ICAM-1, vascular cell adhesion molecule 1 (VCAM 1). They occur, e.g., on the surface of endothelial cells. These connections are characterized by high bond strength. ICAM-1 is a ligand for the CD11a/CD18 integrin. ICAM-1 synthesis is triggered by TNF-α, IL-1 and interferon gamma (IFNγ), endotoxin. ICAM-2 binds to the CD11b/CD18 integrin. The CD11a/CD18 and CD11b/CD18 integrins play the most important role in the inflammatory process [14]. Integrins are also able to bind very different proteins of the extracellular matrix, e.g., fibronectin, fibrinogen, vitronectin, laminin, collagen, plasminogen, osteopontin, von Willebrand factor or sialoprotein of the matrix skeleton [15,16]. The alternative name for β_3_ integrins is cytoadhesins. They play a major role in the adhesion and aggregation of blood platelets and in the formation of complexes. Recognizing the RGD sequence, they bind fibrinogen, vitronectin, fibronectin and von Willebrand factor. They include platelet adhesion gpIIb/IIIa (CD61/CD41) and receptor for vitronectin (CD61/CD51) which occurs on the endothelium and the macrophage cell membrane [17].

Integrin ligands may also include proteolysis-triggered endostatin (coming from type XVIII collagen), endorepellin and tumstatin. Moreover, integrins may also bind viper venom toxins called disintegrins, certain viruses and bacteria [12,18]. Various pathogens, e.g., echoviruses, adenoviruses, and herpesviruses use integrins to penetrate cells. Integrins may be receptors for SARS-CoV-2 and can be implicated in transmission and pathology of SARS-CoV-2 [3].

## 5. Integrin Function

Integrins play an important role in physiological and pathological processes, as well as in wound-healing processes. The specificity of the inflammatory process depends on adequate expression of adhesion molecules enabling leukocyte migration.

During inflammation, integrins enable white blood cells to cross the vascular wall. On the leukocyte membrane, β_2_ integrins bind ICAM-1 whereas α_4_β_1_ and α_4_β_7_ bind VCAM-1 on endothelial cells [19,20]. The integrin α4 subunit can dimerize with either the β_1_ or β_7_ subunit to form the α_4_β_1_ or α_4_β_7_ integrin. During inflammation, α_4_β_1_ promotes transendothelial lymphocyte migration into the inflamed tissue, whereas α_4_β_7_ helps in lymphocyte migration into the intestinal mucosal lymphoid tissues [21]. In addition, the α_4_β_1_ integrin can interact with vascular endothelial growth factor/VEGF receptor 2 (VEGF/VEGFR2) and/or contributes to VEGF functions in chronic lymphocytic leukemia (CLL) [22].

The adhesion and binding of leukocytes to the surface of the vascular endothelium result from the connection of the β_2_ integrin of the leukocyte membrane with endothelial cells. A similar mechanism of adhesion to the vascular endothelium applies to circulating neoplastic cells. These interactions are essential for crossing the vascular barrier and forming metastasis. It has been demonstrated that the occurrence of β_4_ and α_6_ integrins on the cells of squamous cell carcinoma is increased. The CD11b/CD18 integrin (β_2_) mediates responses to Gram-negative bacteria while interleukin 1 takes part in the migration of inflammatory cells. In the case of infection with Gram-positive bacteria, cells migrate via a CD11b/CD18-independent pathway [23,24,25].

Mutations in integrin subunits may cause various genetic diseases in humans. Three autosomal recessive diseases have been described: Glanzmann’s thrombasthenia (mutations of the α_IIb_ and β_3_ integrins), leukocyte adhesion deficiency (LAD)—caused by point mutations or deletion of a gene in the β_2_ integrin—characterized by hereditary deficiency syndrome [26] and vesicular epidermal necrolysis caused by mutation of the α_6_β_4_ integrin [27,28].

## 6. Integrins and the Eye

Integrins play a key role in normal eye development and maintaining tissue homeostasis, and also in the development of pathological processes, such as the healing process of the injuries of the cornea, keratoconus, allergic eye disease, keratitis, dry eye disease, eye infection in the course of COVID-19, lens opacification, glaucoma, diabetic retinopathy, axon degeneration in the optic nerve and scleral remodeling in high myopia (Table 1). Even a slight disorganization of eye tissues due to injury or other pathological disorders may dramatically affect normal vision. Learning about the regulation of integrin activity may serve as an important therapeutic goal.

### 6.1. Integrins and Cornea

The cornea consists of six layers: the epithelium, the epithelial basement membrane (EpBM) (the Bowman’s layer), the stroma, Dua’s layer and the endothelial cell basement membrane (Descemet’s layer) and the endothelium [29]. The acellular layer of the ECM under EpBM is called the Bowman’s layer [30]. The corneal epithelium shows the expression of several integrins. In the central cornea, α_2_β_1_, α_3_β_1_, α_V_β_5_ and α_6_β_4_ integrins are located within the epithelium, with the highest expression level in the basal cells. The α_6_β_4_ integrin mediates adhesion to the EpBM using hemidesmosomes while α_3_β_1_ and α_V_β_5_ involve focal adhesions which are actin-based. The integrins expressed at the EpBM can mediate adhesion of corneal epithelial cells to fibronectins, vitronectin, collagens and laminins. Cells are less proliferative and adhesive to the basement membrane when integrin expression decreases [31].

The studies conducted have shown that in the healing process, skin fibroblasts use the α_2_ and β_1_ integrins and the integrin αv subunit. Removal of the β_1_ integrin delays closure of skin wound [32]. In the case of corneal wound healing, neutrophils and lymphocytes are released from damaged corneal epithelium and then reach the injured site [33]. Migration of neutrophils depends on integrins. They migrate along the keratocyte network. These active keratocytes densify the region of stromal injuries [34]. The interactions between keratocytes/myofibroblasts and their integrins are involved in wound healing, and also in fibrosis which causes opacification of the cornea that may decrease visual acuity [35,36]. Fibrosis impacts the differentiation of stromal keratocytes into stress fibers containing myofibroblasts [37]. Stress fibers are formed by α-smooth muscle actin (SMA), a specific marker of myofibroblasts that arrest normal transmission of light through the stroma and disturb vision [38]. The integrin α_V_β_6_ is upregulated on injury [39] and is able to induce fibrosis as knocking out α_V_β_6_ was found to decrease the level of SMA and thrombospondin 1 (TSP-1) activating TGF-β at an early stage of the wound-healing process. This process may play an important role in the treatment of scars of the cornea [40]. In vitro studies revealed an increase in TGF-β1 signaling and accelerated terminal differentiation of keratinocytes in mice [41]. The α_6_-encoding gene mutations and β proteins expressed in the hemidesmosome lead to epidermolysis bullosa that causes ocular surface changes [42].

In addition, a recent study by Lin et al. has demonstrated that an extracellular matrix protein, transforming growth factor β-induced protein (TGFBIp), is increased in inflamed mouse corneas and may lead to lymphatic sprouting in corneal lymphangiogenesis (LA). The TGFBIp upregulation in a sutured cornea is correlated with infiltration of macrophages. TGFBIp stimulates migration, tube formation and adhesion of human lymphatic endothelial cells (HLECs) but has no impact on the proliferation of HLECs. In vitro, the TGFBIp’s effect is mediated by the pathway of integrin α_5_β_1_-FAK. Integrin α_5_β_1_ blockade can significantly inhibit lymphatic sprouting induced by TGFBIp [43].

It should be noted that abnormal corneal LG is observed in several ocular diseases, e.g., in dry eye disease, transplant rejection, herpetic keratitis and eye allergy.

The interest in the α_6_β_4_ increase revealed that patients who have cicatricial pemphigoid possess anti-integrin β_4_ autoantibodies that bind to the basal surface of epithelial cells in the cornea and can lead to severe damage to the ocular surface and inflammation [44].

Weller et al. found out upregulation of integrins α_1_, α_3_, α_4_, α_L_, β_1_, β_3_ and β_4_ in explanted endothelial cell-DM specimens from Fuchs’ patients [45]. Fuchs’ corneal dystrophy is a hereditary eye disease in which corneal endothelial degeneration and underlying Descemet membrane (DM) lead to corneal edema and haze. Fuchs’ progression is characterized by DM thickening that is associated by increased EMT by the endothelial layer and ECM deposit formation. Further research is needed to assess the expression of integrins in the Fuchs’ cornea to discover the impact of interactions between the corneal endothelium and DM on the pathogenesis and progression of this pathology.

### 6.2. Integrins and Allergic Eye Diseases

Integrins play an important role in allergic eye diseases, such as spring conjunctivitis and keratitis, atopic conjunctivitis and keratitis and giant papillary conjunctivitis. These diseases manifest themselves with lacrimation, itching, redness, burning and photophobia. An allergic reaction triggers release of mediators which cause migration of inflammatory cells (eosinophils, neutrophils, basophils and lymphocytes) to the reaction site. Cell adhesion molecules (selectins, integrins and immunoglobulins) are also part thereof. The first stage involves activation of P- and E-selectins, integrins, such as α_4_β_1_, α_4_β_7_, α_M_β_2_, α_L_β_2_, and immunoglobulins (ICAM-1, ICAM-2, VCAM-1, VCAM-4, LFA-1), which ensure tight bond of eosinophils and basophils with vascular endothelial cells [46,47].

Integrins may affect keratitis which is a finding of epidemic keratoconjunctivitis (EKC) induced by six adenoviruses HAdV (HAdV-8, -19, -37, -53, -54 and -56). HAdV-37 interacts with integrin α_V_β_5_ to enter non-ocular human cells. Storm et al. failed to find α_V_β_5_ expression on human corneal epithelial cells and showed that integrins α_V_β_1_ and α_3_β_1_ are important for HAdV-37 infection of the corneal tissue [48].

### 6.3. Integrins and Dry Eye Disease

The prevalence of dry eye disease (DED) is high. DED is characterized by visual disturbance, symptoms of discomfort and tear film instability. It has been known that inflammation of the ocular surface and lacrimal gland is crucial in this disease [49]. The activation of T cells contributes to the inflammatory process and involves aggregation of several cell surface proteins between the antigen-presenting cells and T cells, including binding of the integrin LFA-1 to its cognate ligand, intercellular adhesion molecule 1 (ICAM-1). Importantly, LFA-1 and ICAM-1 interact in T cell adhesion and migration at inflammation sites. In recent years, numerous studies have been conducted on the regional use of the LFA-1 antagonist in dry eye syndrome. The studies revealed that the regional use of anti- LFA-1 led to significant improvement of symptoms of dry eye syndrome and suppression of inflammatory processes [50].

### 6.4. Integrins and Eye Infection in the Course of COVID-19

In 2019, there was the first report about a new viral infection in Wuhan, China. The new virus was designated as severe acute respiratory syndrome coronavirus 2 (SARS-CoV-2) that causes coronavirus disease 2019 (COVID-19). SARS-CoV-2 uses the angiotensin-converting enzyme 2 (ACE2) receptor for cell invasion, which is expressed in different tissues such as the lungs, the small intestine, the testicles, the kidneys, the brain and the eye [51]. ACE2 is the main cell surface receptor of SARS-CoV-2 that binds the virus’ spike protein. Postmortem analysis of the eyes of patients with COVID-19 showed expression of ACE2 in the conjunctiva, limbus and cornea, especially in the superficial conjunctival and corneal epithelial surface. Surgical conjunctival specimens also revealed expression of ACE2 in the conjunctival epithelium, especially significant in the superficial epithelium. All postmortem eyes and surgical specimens also expressed transmembrane protease, serine 2 (TMPRSS2), a cell surface-associated enzyme that facilitates viral entry following binding of the viral spike protein to ACE2 [52]. These results suggest that ocular surface cells can be susceptible to infection by SARS-CoV-2. The study by Collin et al. indicates that local inflammation can enhance ACE2 expression, which explains overexpression of ACE2 in diseased conjunctiva. SARS-CoV-2 can also use integrins as cell receptors, binding to them through the RGD motif which is present in the receptor-binding domain in the spike protein of SARS-CoV-2 [53]. The RGD motif is the minimal peptide sequence required for binding proteins of the integrin family [54] that is involved in the binding of human ACE2 [55]. Viral proteins with RGD motifs promote infection by binding integrin heterodimers including α_V_β_1_, α_V_β_3_, α_V_β_5_, α_V_β_6_, α_V_β_8_, α_5_β_1_, α_8_β_1_ and α_IIb_β_3_ [54] activating transducing pathways involving phosphatidylinositol-3 kinase (PI-3K) or mitogen-activated protein kinase (MAPK), which promote virus entry and infection of the host cell [3]. Patients with COVID-19 may present ocular manifestations such as conjunctival congestion, chemosis, increased tearing, ocular pain and foreign body sensation [56,57].

### 6.5. Integrins and the Lens

Studies of the lens observed that the α_3_β_1_, α_6_β_1_, α_6_β_4_, α_1_β_1_, α_2_β_1_, α_5_β_1_ and α_V_β_3_ integrins are expressed in the developing lens [58,59]. Altered integrin function is connected with the progression of fibrotic diseases of the lens. The α_V_β_5_, α_V_β_6_ and α_V_β_3_ integrins play a role in the regulation of transforming growth factor β (TGF-β) signaling pathways in posterior capsule opacification (PCO) and anterior subcapsular cataract (ASC). The regulation of TGF-β signaling may prevent lens epithelial-to-mesenchymal transition (EMT) which leads to cataract [60,61].

### 6.6. Integrins and Glaucoma

Glaucoma is a heterogeneous disease with alterations in the trabecular meshwork (TM) and the optic nerve head (ONH) [62]. The glaucoma-related phenotypic changes involve the actomyosin-based contractile properties of TM, ECM compliance and the types and quantity of proteins present in the ECM in the TM and the ONH. Integrin receptors are engaged in this bidirectional communication [63,64].

The α_V_β_3_ integrin plays a crucial role in the pathomechanism of glaucoma. The level of the α_V_ integrin was elevated in the layer of retinal ganglion cells (RGC) and in the glial cells of the nerve head after optic nerve crush in mice [65]. A study by Dickerson et al. [66] revealed downregulation of the α_V_ integrin unit expression by glucocorticoids. Other studies found an increase in the β_3_ integrin unit expression in the TM [63,67]. The osteopontin-activated α_V_β_3_ integrin signaling has been shown to control the eNOS/NO pathway that is involved in the outflow facility [68,69].

### 6.7. Integrins and the Retina

Brem et al. studied expression of integrins from three different subfamilies in the human retina. Nine subunits showed unique distribution in the eye. Understanding the distribution of cell adhesion molecules in the retina will help to identify their function in the eye [70]. ICAM-1 is an adhesion molecule on endothelial cells and leukocytes which participates in the recruitment of leukocytes and inflammatory sites [71]. Increased levels of ICAM-1 expression occur in diabetic retinopathy [72]. A chronic inflammatory process is an important link in the development of type 2 diabetes. Leukocytes play an important role in the course of inflammatory processes at each stage of diabetes development, and also in diabetes complications in the form of micro- and macroangiopathy. As mentioned before, on the surface of leukocytes, there occur β_2_ integrins which participate in adhesion to vascular walls by binding the ICAM-1 molecule situated on the endothelium. Activation of leukocytes leads to increased expression of β_2_ integrins. Therefore, it is believed that the level of β_2_ integrin expression reflects the state of leukocyte activation. Enhanced leukocyte adhesion to vascular walls and production of factors with a potential cytotoxic effect (free radicals, TNF-α) that destroy the vascular membrane (free radicals, metalloproteinases) and activate inflammatory response of vascular walls (IL-1, TNF-α, arachidonic acid derivatives) and fibroblast proliferation may lead to accelerated development of atheromatous lesions and damage to small vessels [73]. Some studies indicate increased expression of β_2_ integrins on leukocytes in diabetic patients [74]. The integrin beta subunit CD18 is increased in patients with diabetic retinopathy (DR) [75], and likewise significant increases in integrin alpha subunits CD11a [76] and CD11b [73] are found in these patients.

Retinopathy of prematurity (ROP), a two-phase disease of retinal vascular development, is caused by VEGF deregulation [77,78]. VEGF is downregulated in infants exposed to hyperoxia in phase 1, whereas in phase 2, VEGF is upregulated in relative/true hypoxia. There are several isoforms of VEGF, among which VEGFA165 predominates in the eye with multiple pro- and antiangiogenic splice variants [79]. A study conducted using a newborn mouse model of oxygen-induced retinopathy (OIR) showed that oxidative stress from fluctuating hyperoxia and hypoxia results in abnormal vascular development resembling human ROP [77]. An experimental study by Wilkinson-Berka et al. showed that the antagonist of the α_V_β_3_ integrin attenuates angiogenesis in ROP and increases the overexpression of VEGF [80].

Within the adult central nervous system (CNS), tenascin C (TN-C) is the major ECM glycoprotein, upregulated at sites of damage in the brain, spinal cord and optic nerve. The integrin enabling cells to migrate on TN-C, α_9_β_1_ is present in the CNS in the embryonic growth. Then, it is downregulated in adulthood and is not upregulated after injury. Integrins are transported into retinal ganglion cell axons in the retina, although this may be limited in the optic nerve, with the axons containing expressed integrins. Transduction of ganglion cells with the α_9_ integrin and kindlin-1 promotes regeneration of these axons, but the transport may be needed for regeneration of the remaining axons [81].

### 6.8. Integrins and Myopia

Integrins may impact scleral remodeling in high myopia leading to biomechanical weakening and continued scleral creep. An experimental study reported diminished levels of collagen-binding integrins α_1_ and β_1_ during myopia progression [82]. Additionally, it was observed that basic fibroblast growth factor (bFGF) inhibited form deprivation myopia (FDM) in the experimental model; however, it was found to upregulate the levels of type 1 collagen, α_2_ integrin and β_1_ integrin [83]. The latest research found all the known integrin alpha subunits except α_D_ and α_E_ in the scleral tissue of guinea pigs [84].

### 6.9. Future Directions and Anti-Integrin Therapy

The pathogenesis of many eye disorders characterized by pathological angiogenesis and inflammation involves deregulation of integrins. In these diseases, therapeutic procedures may directly target integrins or their ligands.

Recent investigations have emphasized an important role of the α_4_ integrin in allergic conjunctivitis and indicated a potential treatment of local allergy using DS-70, a novel α/β-peptidomimetic α_4_ integrin antagonist, to inhibit the conjunctival infiltration of immune cells [85].

Integrin antagonists can improve signs and symptoms of patients suffering from dry eye disease, posterior capsular opacification, age-related macular degeneration, diabetic macular edema and vitreomacular traction. In the anterior segment, anti-integrin therapy can be employed to effectively treat DED. In this approach, lifitegrast, a small-molecule integrin antagonist, binds the CD11a subunit of LFA-1 and blocks the interaction between LFA-1 and ICAM-1. In this way, it blocks the inflammatory pathways and inhibits subsequent T cell-mediated inflammation in DED [86,87]. An experimental study by Ecoiffier et al. demonstrated that local application of an anti-very late antigen 4 small-molecule antagonist (anti-VLA-4 sm) anti-integrin α_4_β_1_ caused a reduction in dry eye signs and suppression of inflammatory changes such as a significant decrease in conjunctival T cell numbers and TNF-α transcript levels in the cornea and conjunctiva [46]. Local blockade of VLA-4 may be a new therapeutic approach in the treatment of clinical signs and inflammatory alterations connected with DED.

Therapeutic agents which are α_V_ integrin antagonists may be used to prevent posterior capsular opacification after cataract removal surgery. The research on mice revealed that cells deprived of the α_V_ integrin were transparent and showed no abnormalities [88].

Integrin peptide therapy for a posterior segment pathology significantly affects neovascular retinal diseases. This therapy targets integrin receptors that take part in cell signaling, regulation and construction of new and aberrant blood vessels and improves vision by regressing and inhibiting new blood vessel formation and by reducing retinal vascular leakage in the retina. Numerous clinical trials have been conducted to evaluate the efficacy of integrin antagonist therapy for age macular degeneration (AMD), proliferative diabetic retinopathy (PDR), macular edema, vitreomacular adhesion and traction. ALG-1001 is a novel α_V_β_3_, α_V_β_5_ and α_5_β_1_ integrin inhibitor which has been demonstrated to affect both vitreolysis and angiogenesis. In the retina, integrin receptors α_V_β_3_, α_V_β_5_ and α_5_β_1_ are associated with angiogenesis, both with choroidal angiogenesis found in dry and wet AMD and with preretinal angiogenesis connected with diabetic macular edema (DME) and retinal vein occlusion [89,90,91]. A phase IIb clinical trial assessing the safety and efficacy of ALG-1001 in patients with vitreomacular adhesion (VMA) and vitreomacular traction (VMT) revealed good tolerance and lack of intraocular inflammation [92].

Integrin-targeted therapy is promising in ROP. Targeting α_2_β_1_ integrin expression on endothelial cells reduces oxygen-induced retinopathy (OIR) [93]. The application of a non-peptide antagonist of the α_V_β_3_ integrin, i.e., of 3-(3-(6-guanidino-1-oxoisoindolin-2-yl)propanamido)-3-(pyridin-3yl)propanoic acid dihydrochloride, may inhibit retinal neovascularization [94]. There are interesting possibilities that endothelial α_2_β_1_ may be a therapeutic target in pathological angiogenesis.

Summing up, the regulation of integrin expression is difficult since each cell adhesion molecule covers more than one type of cells and has several ligands which in combination result in various effects. In order to fully understand complex adhesion processes, it is necessary to conduct further studies and to search for factors that affect integrin expression. Thorough knowledge of the activity of integrins may contribute to the development of new diagnostic methods and immunotherapy of eye diseases.

## Figures and Tables

**Table 1 cells-10-01703-t001:** Integrins involved in the normal eye and eye diseases.

Part of the Eye/Disease	Integrins
Cornea/	
normal expression	α_2_β_1_, α_3_β_1_, α_V_β_5_, α_6_β_4_
injury	α_V_β_6_
healing process of the injury	α_2,_ β_1_
corneal lymphangiogenesis	α_5_β_1_
cicatricial pemphigoid	α_6_β_4_
epidemic keratoconjunctivitis	α_V_β_1_, α_V_β_5_, α_3_β_1_
Conjunctiva and cornea/	
allergic conjunctivitis	α_4_β_1_, α_4_β_7_, α_M_β_2_, α_L_β_2_
epidemic keratoconjunctivitis	α_V_β_1_, α_V_β_5_, α_3_β_1_
dry eye disease	α_L_β_2_
COVID-19	α_V_β_1_, α_V_β_3_, α_V_β_5_, α_V_β_6_, α_V_β_8_, α_5_β_1_, α_8_β_1_, α_IIb_β_3_
Lens/	
normal development	α_3_β_1_, α_6_β_1_, α_6_β_4_, α_1_β_1_, α_2_β_1_, α_5_β_1_, α_V_β_3_
Cataract	α_V_β_5_, α_V_β_6_, α_V_β_3_
Trabecular meshwork or optic nerve head/	
Glaucoma	α_V_β_3_
Retina/	
diabetic retinopathy	α_L_β_2_
retinopathy of prematurity	α_V_β_3_
regeneration of the axons of the optic nerve	α_9_β_1_
sclera/	
myopia	α_1_, β_1_

## Data Availability

Not applicable.

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
