# Peer review of "Integrins: An Important Link between Angiogenesis, Inflammation and Eye Diseases"

_cells, 2021, doi:10.3390/cells10071703_

Round 1
Reviewer 1 Report
The review “Integrins: A missing link between angiogenesis, inflammation and eye diseases” is a comprehensive on the topic. I just have a few minor comments as delineated below:
- Line 111: please correct to RGD. “Integrins recognizing RGD”
- Figure 1: please change title to “Leukocyte-specific receptors” to match spelling of leukocyte throughout rest of text.
- Spaces needed: line 226 space between and β proteins
Line 345 space between by pathological
- Line 392 it should say “integrins”
- Line 321 needs correction. As stated “VEGF exposed to hyperoxia is down regulated in Phase 1” is not correct. Perhaps “ VEGF is down regulated in infants exposed to hyperoxia” but certainly VEGF is not exposed to hyperoxia. This needs to be corrected.
Author Response
Response to Reviewer 1 Comments
Thank you for preparing the review of our manuscript ID: cells- 1213134. All your comments were taken into account.
Point 1.
Line 111: please correct to RGD. “Integrins recognizing RGD”
Response 2.
Thank you for your remark. It was changed to “RGD”
Point 2.
Figure 1: please change title to “Leukocyte-specific receptors” to match spelling of leukocyte throughout rest of text.
Response 2.
Thank you for your remark. We have made appropriate changes to “Leukocyte-specific receptors”.
Point 3.
Spaces needed: line 226 space between and β proteins
Line 345 space between by pathological
Response 3.
Thank you for your remark. We have made the spaces: and βproteins, by pathological
Point 4.
Line 392 it should say “integrins”
Response 4.
It has been corrected as “ integrins”
Point 5.
Line 321 needs correction. As stated “VEGF exposed to hyperoxia is down regulated in Phase 1” is not correct. Perhaps “ VEGF is down regulated in infants exposed to hyperoxia” but certainly VEGF is not exposed to hyperoxia. This needs to be corrected.
Response 5.
Thank you for your remark. This part of the sentence was corrected.
VEGF is down regulated in infants exposed to hyperoxia …

Reviewer 2 Report
The paper of Mrugacz et al. is a review on the Integrins, starting from a general introduction on their structure, activation and classification to take the discussion on their role in eye diseases and on the possibility they may offer as therapeutic targets. Even if the review does not cover the whole literature of the topic, it offers a picture of the complexity of the role of integrins in health eye tissues and in the development of diseases. For this reason, I suggest the paper for the publication on Cells after some revisions.
The English could be improved above all in its fluidity. Some phrases are not completely clear and I suggest a revision by a mother-tongue.
Revision:
Figure 1 is taken from Ref.1. It should be modified because of copyright.
Lines 111 and 167 RGD in place of RGT
In the paragraph “Integrin function” the authors speak about the role of β2 integrins on the leukocyte membrane in the binding of ICAM-1 and VCAM-1 on endothelial cells during inflammation. β2 integrins bind ICAM-1 and not VCAM-1. However, α4β1 and α4β7 bind VCAM-1 and during inflammation play a role analogous to β2 integrins, but these integrins are not even cited in this paragraph. This paragraph should be improved inserted the role of α4β1 and α4β7 in the inflammation process.
Line 116: p150,95 in place of 150.95
Line 122: ICAM-1 in place of 1CAM-1
Line 136, the phrase “This is a group with a common β2 chain, also called CD18.” The part “also called CD18”, can be removed since it was already said in line 114. The first part of the phrase, “This is a group with a common β2 chain”, is not very clear to me. Maybe it could be removed, and the next sentence can begin with: “This subunit can be linked….ect”
Line 140: Mac-1 in place of MAC-1
Line 144: “Integrin ligands also include opposite receptors”. What does opposite mean referred to receptors?
Line 154: “With the participation of RGD complex” could be changed in “Recognizing the RGD sequence,”
Line 167, change “triple amino acid sequence” with “tri-amino acid sequence”
Line 200: I suggest to remove (the Bowman’s layer), because it is put in brackets after EpBM and seems a synonym of EpBM, but two lines after the authors explain that “The Browman’s layer refers not to the EpBM but to the acellular layer of ECM under EpBM.” This phrase could be changed in “The acellular layer of ECM under EpBM is called the Browman’s layer”, or something similar.
In line 243 the acronym DM is used, but it was not defined, as well as ROP in line 320.
Line 359-360: two brackets are open but only one is close.
Line 384: I suggest to remove “novel”. The compound was novel in 2014 when it was published
There are several missing spaces between words in the text, for example in line 125 “β7chains”. I found this kind of mistakes in lines 125, 128, 133, 180, 226, 240, 244, 281, 343, 345, 346 and 383. Please, check carefully all the text.
The authors should uniform the way of writing the integrins. In some cases the numbers are subscript, in others the numbers are in line with the Greek letter. In line 329 and 357 the authors use the whole words Alpha9beta1 and alpha4beta1.
Author Response
Response to Reviewer 2 Comments
Thank you for preparing the review of our manuscript ID: cells-1213134. All your comments were taken into account.
Point 1.
Figure 1 is taken from Ref.1. It should be modified because of copyright.
Response 1:
Thank you for your remark. It was added Reference No 1 [1] at the end of the description of Fig. 1.
Point 2.
Lines 111 and 167 RGD in place of RGT
Response 2.
Thank you for your remark. We have made appropriate changes in the point.
Point 3.
In the paragraph “Integrin function” the authors speak about the role of β2 integrins on the leukocyte membrane in the binding of ICAM-1 and VCAM-1 on endothelial cells during inflammation. β2 integrins bind ICAM-1 and not VCAM-1. However, α4β1 and α4β7 bind VCAM-1 and during inflammation play a role analogous to β2 integrins, but these integrins are not even cited in this paragraph. This paragraph should be improved inserted the role of α4β1 and α4β7 in the inflammation process.
Response 3.
Thank you for your remark. We have made the correction in the point.
During inflammation, integrins enable white blood cells to cross the vascular wall. β2 integrins on the leukocyte membrane bind ICAM-1 whereas α4β1 and α4β7 bind VCAM-1 on endothelial cells [17-18]. The integrin α4 subunit can dimerize with either β1or β7 subunit to form α4β1 and α4β7-integrin. During inflammation, α4β1 promotes lymphocyte trans-endothelial migration into inflamed tissue, whereas α4β7 helps in the lymphocyte migration into the intestinal mucosal lymphoid tissues [19]. In addition, α4β1 integrin can interact with vascular endothelial growth factor/ VEGF receptor 2 (VEGF/VEGFR2) or/and contributes to VEGF functions in chronic lymphocytic leukemia (CLL) [20].
Point 4.
Line 116: p150,95 in place of 150.95
Response 4.
It has been corrected as “150,95”
Point 5.
Line 122: ICAM-1 in place of 1CAM-1
Response 5.
Thank you for your remark. It has been corrected as “ICAM-1”
Point 6.
Line 136, the phrase “This is a group with a common β2 chain, also called CD18.” The part “also called CD18”, can be removed since it was already said in line 114. The first part of the phrase, “This is a group with a common β2 chain”, is not very clear to me. Maybe it could be removed, and the next sentence can begin with: “This subunit can be linked….ect”
Response 6.
Thank you for your remark. The phrase “This is a group with a common β2 chain, also called CD18.” was removed and next sentence was changed.
„This subunit can be linked with one of three α subunits which form the CD11a, CD11b, CD11c titre”.
Point 7.
Line 140: Mac-1 in place of MAC-1
Response 7.
Thank you for your remark. This point has been corrected to ‘Mac-1”
Point 8.
Line 144: “Integrin ligands also include opposite receptors”. What does opposite mean referred to receptors?
Response 8.
Thank you for your remark. I apologize for this „strange” error. I corrected this sentence to „Integrin ligands also include receptors, which belong to a family of immunoglobulin-like CAMs such as ICAM-1, vascular cell adhesion molecule-1 (VCAM 1). They occur e.g., on the surface of endothelial cells”.
Point 9.
Line 154: “With the participation of RGD complex” could be changed in “Recognizing the RGD sequence,”
Response 9.
Thank you for your remark. It was changed in “Recognizing the RGD sequence,”
Point 10.
Line 167, change “triple amino acid sequence” with “tri-amino acid sequence”
Response 10.
Thank you for your remark. It was changed in “ tri-amino acid sequence”
Point 11.
Line 200: I suggest to remove (the Bowman’s layer), because it is put in brackets after EpBM and seems a synonym of EpBM, but two lines after the authors explain that “The Browman’s layer refers not to the EpBM but to the acellular layer of ECM under EpBM.” This phrase could be changed in “The acellular layer of ECM under EpBM is called the Browman’s layer”, or something similar.
Response 11.
Thank you for your remark. It was changed in “The acellular layer of ECM under EpBM is called the Browman’s layer”.
Point 12.
In line 243 the acronym DM is used, but it was not defined, as well as ROP in line 320.
Response 12.
Thank you for your remark. Acronyms were inserted to: Descemet membrane (DM) and Retinopathy of prematurity (ROP).
Point 13.
Line 359-360: two brackets are open but only one is close.
Response 13.
Thank you for your remark. It was changed to „Experimental study by Ecoiffier et al. demonstrated that local application of anti-very late antigen 4 small-molecule antagonist (anti-VLA-4 sm), which is anti-integrin α4β1, caused a reduction in dry eye signs and suppression of inflammatory changes such as a significant decrease in conjunctival T-cell numbers and tumor necrosis factor-alpha transcript levels in the cornea and conjunctiva”
Point 14.
Line 384: I suggest to remove “novel”. The compound was novel in 2014 when it was published
Response 14.
You are absolutely right. „Novel” was removed.
Point 15.
There are several missing spaces between words in the text, for example in line 125 “β7chains”. I found this kind of mistakes in lines 125, 128, 133, 180, 226, 240, 244, 281, 343, 345, 346 and 383. Please, check carefully all the text.
Response 15.
Thank you for your remark. We removed these missing spaces.
Point 16.
The authors should uniform the way of writing the integrins. In some cases the numbers are subscript, in others the numbers are in line with the Greek letter. In line 329 and 357 the authors use the whole words Alpha9beta1 and alpha4beta1.
Response 16.
Thank you for your remark. It is now uniform the way of writing the integrins.
Reviewer 3 Report
The following corrections are required:
#1
The authors mention SARS-COV2 several times in both the abstract and text. Do the authors intend to discuss the relationship between SARS-COV2 and eye infection in the manuscript?
This relationship is very interesting; however, the authors’ description of SARS is inadequate. Additionally, there are already several hypotheses regarding the relationship between the SARS-COV2 and eye infection.
The authors should either mention in the text that RGD motif and integrin binding may be involved in eye infection (PMID: 32130973) or similar to other authors, mention in the text that there is a relationship between ACE2 and the virus. If the latter is mentioned, then it should be noted that ACE2 is expressed in the human corneal and conjunctival tissues (PMID: 32289466).
#2
There are multiple integrins listed in the text.
Please summarize the integrins involved in each disease or expressed in each organ. Creating a new table will facilitate the readers’ understanding of the multiple integrins.
#3
In line 111 and 167, RGT > RGD
#4
In line 112, please include α8β1 in RDG integrin, consistent with Figure 1.
#5
In line 168, α6β4 > α6β4)
#6
In line 156, gpIIb\IIIa > gpIIb/IIIa
#7
In line 181, 182, G- or G+ should be spelled out to gram negative or gram positive.
Author Response
Response to Reviewer 3 Comments
Thank you for preparing the review of our manuscript ID: Cells-1213134. All your comments were taken into account.
Point 1.
The authors mention SARS-COV2 several times in both the abstract and text. Do the authors intend to discuss the relationship between SARS-COV2 and eye infection in the manuscript?
This relationship is very interesting; however, the authors’ description of SARS is inadequate. Additionally, there are already several hypotheses regarding the relationship between the SARS-COV2 and eye infection.
The authors should either mention in the text that RGD motif and integrin binding may be involved in eye infection (PMID: 32130973) or similar to other authors, mention in the text that there is a relationship between ACE2 and the virus. If the latter is mentioned, then it should be noted that ACE2 is expressed in the human corneal and conjunctival tissues (PMID: 32289466).
Response 1:
Thank you for your remark. We have made the correction in the point.
6.4. Integrins and eye infection in the course of COVID-19
In 2019 there was the first report about a new viral infection in Wuhan, China. The new virus was designed as Severe Acute Respiratory Syndrome Coronavirus 2 (SARS-CoV-2) that causes the coronavirus disease 2019 (COVID-19). SARS-CoV-2 uses the angiotensin-converting enzyme 2 (ACE2) receptor for cell invasion, which is expressed in different tissues such as lungs, small intestine, testicles, kidneys, brain, and the eye [49]. ACE2 is the main cell-surface receptor for SARS-CoV-2 that binds the viral spike protein. Analysis post-mortem eyes of patients with COVID-19 showed expression of ACE2 in the conjunctiva, limbus, and cornea, especially in the superficial conjunctival and corneal epithelial surface. Surgical conjunctival specimens also revealed expression of ACE2 in the conjunctival epithelium, especially significant in the superficial epithelium. All post-mortem eyes and surgical specimens also expressed Transmembrane protease, serine 2 (TMPRSS2), a cell surface-associated enzym that facilitates viral entry following binding of the viral spike protein to ACE2 [50]. These results suggest that ocular surface cells can be susceptible to infection by SARS-CoV-2. The study by Collin et al. indicates that local inflammation can enhance ACE2 expression, which explains overexpression of ACE2 in diseased conjunctiva SARS-CoV-2 can also use integrins as cell receptors, binding to them through a RGD motif which is present in the receptor-binding in the spike protein of SARS-CoV-2 [51]. The RGD motif is the minimal peptide sequence required for binding proteins of the integrin family [52] that is involved in the binding of human ACE2 [53]. Viral proteins with RGD motifs promote infection by binding integrin heterodimers including αVβ1, αVβ3, αVβ5, αVβ6, αVβ8, α5β1, α8β1 and αIIbβ3 [54] activating transducing pathways involving phosphatidylinositol-3 kinase (PI–3K) or mitogen-activate protein kinase (MAPK), which promote virus entry and infection of the host cell [3]. Patients with COVID-19 may present ocular manifestations such as conjunctival congestion, chemosis, increased tearing, ocular pain and foreign body sensation [55-56].
Point 2.
There are multiple integrins listed in the text.
Please summarize the integrins involved in each disease or expressed in each organ. Creating a new table will facilitate the readers’ understanding of the multiple integrins.
Response 2.
Thank you for your remark. We created a new table (Table 1).
|
Part of the eye/ disease |
Integrins |
|
Cornea/ normal expression injury the healing process of the injury corneal lymphangiogenesis cicatricial pemphigoid |
α2β1, α3β1, αvβ5, α6β4 αvβ6 α2, β1 α5β1 α6β4 |
|
Conjunctiva/and cornea/ allergic conjunctivitis epidemic keratoconjunctivitis dry eye disease COVID-19 |
αM αvβ1, αvβ5, α3β1 LFA-1 αVβ1, αVβ3, αVβ5, αVβ6, αVβ8, α5β1, α8β1, αIIbβ3 |
|
Lens/ microphthalmia cataract |
β1 αv |
|
Trabecular meshwork or optic nerve head/ glaucoma |
αvβ3 |
|
Retina/ diabetic retinopathy retinopathy of prematurity regeneration of the axons of the optic nerve
Sclera/ Myopia |
β2 αvβ3 α9β1
α1, β1 |
Table 1. Integrins involved in normal eye and eye diseases
Point 3.
In line 111 and 167, RGT > RGD
Response 3.
Thank you for your remark. We have made the correction in the point.
Point 4.
In line 112, please include α8β1 in RDG integrin, consistent with Figure 1.
Response 4.
Thank you for your remark. It has been added.
Point 5.
In line 168, α6β4 > α6β4)
Response 5.
Thank you for your remark. It was corrected.
Point 6.
In line 156, gpIIb\IIIa > gpIIb/IIIa
Response 6.
Thank you for your remark. We have made the correction in the point.
Point 7.
In line 181, 182, G- or G+ should be spelled out to gram negative or gram positive.
Response 7.
Thank you for your remark. We have made proper changes.
In the case of infection with gram positive bacteria, cells migrate on a CD11b/CD18-independent pathway [28–31].
Reviewer 4 Report
The review-manuscript by Mrugacz, Bryl, Falkowski, and Zorena had as the main focus to show the role of integrins in the development and physiology and, mainly, in the pathophysiology of different eye diseases, emphasizing these adhesion receptors as potential therapeutic targets.
However, although the main focus of the work is Integrinas and Eye Diseases, which is a very interesting topic that gives ample possibility to discuss what is in the literature and the pharmacological and therapeutic novelties, serious flaws in the structure of the review, weaken the work.
This reviewer believes that the topic is of great interest and the manuscript should be deeply restructured and revised at several points in order to be published and reach the target audience of Cells:
Some main issues are addressed below:
1- Title: The statement that integrins would be the “missing link" in angiogenesis, inflammation, and eye diseases, does not seem rather correct. Although is understood that the authors wanted to call attention to integrins in eye diseases as a therapeutic target, the role of these adhesion receptors in these phenomena, which occur in eye disease, are well known.
2- In a review work, like this one, the use of other authors' reviews as references should be avoided, or used only when necessary. This is not the case here, since more than 30% of the references are older reviews on the subjects covered.
3- The initial structure of the texts that make up the manuscript does not help a better understanding of the main point to be discussed (the role of Integrins in eye diseases). The items, from 1 (Introduction) up to 5 (Integrins ... structure, activation, classification, and functions) contain a lot of information but presented rather superficially. In an attempt to address various points about integrins, the authors used an excessive number of literature reviews, there are some loose sentences and sometimes truncated. The text is rather extensive and for the target readers of Cells, it contains excessive basic information (Taking into account the necessary data for a better understanding of the main subject - Integrin and eyes).
In addition, the text, as well as in the entire manuscript, has a large number of phrases in plagiarism taken from various works (Checked at Plagiarisma online); Including Figure 1, which is the same as that presented in a recently published article (https://doi.org/10.1038/s41390-020-01177-9).
I suggest that the introduction be shortened, containing the essential and necessary information for introducing the following item (6), avoid repeating the same information of the following items.
4- Item 6: contains the mainline of the study and is well structured, with relevant and interesting information, although in items 6.3; 6.4, and 6.5, the issues could be further explored. The discussion on the use of integrin blockers/antagonists in the treatment of eye diseases is a highlighted point that also deserves to be better discussed.
However, the whole text in item 6 needs to be thoroughly revised since several sentences are exact copies of the main articles cited and, in some points, mixed sentences make it difficult to understand, as they do not connect.
Author Response
Response to Reviewer 4 Comments
Thank you for preparing the review of our manuscript ID: cells-1213134. All your comments were taken into account.
Point 1.
Title: The statement that integrins would be the “missing link" in angiogenesis, inflammation, and eye diseases, does not seem rather correct. Although is understood that the authors wanted to call attention to integrins in eye diseases as a therapeutic target, the role of these adhesion receptors in these phenomena, which occur in eye disease, are well known.
Response 1:
Thank you for your feedback. It was changed to “Integrins: An Important Link between Angiogenesis, Inflammation and Eye Diseases”
Point 2.
In a review work, like this one, the use of other authors' reviews as references should be avoided, or used only when necessary. This is not the case here, since more than 30% of the references are older reviews on the subjects covered.
Response 2.
Thank you for your remark. We have made appropriate changes to the point. I removed 16 old references.
Point 3.
The initial structure of the texts that make up the manuscript does not help a better understanding of the main point to be discussed (the role of Integrins in eye diseases). The items, from 1 (Introduction) up to 5 (Integrins ... structure, activation, classification, and functions) contain a lot of information but presented rather superficially. In an attempt to address various points about integrins, the authors used an excessive number of literature reviews, there are some loose sentences and sometimes truncated. The text is rather extensive and for the target readers of Cells, it contains excessive basic information (Taking into account the necessary data for a better understanding of the main subject - Integrin and eyes).
In addition, the text, as well as in the entire manuscript, has a large number of phrases in plagiarism taken from various works (Checked at Plagiarisma online); Including Figure 1, which is the same as that presented in a recently published article (https://doi.org/10.1038/s41390-020-01177-9).
I suggest that the introduction be shortened, containing the essential and necessary information for introducing the following item (6), avoid repeating the same information of the following items.
Response 3.
Thank you for your remark. We have made the correction to the point.
We carefully checked our manuscript. The entire work was corrected by the native speaker from our University. Regarding plagiarism- it was checked using MDPI System.
Figure 1was removed.
Point 4.
Item 6: contains the mainline of the study and is well structured, with relevant and interesting information, although in items 6.3; 6.4, and 6.5, the issues could be further explored. The discussion on the use of integrin blockers/antagonists in the treatment of eye diseases is a highlighted point that also deserves to be better discussed.
However, the whole text in item 6 needs to be thoroughly revised since several sentences are exact copies of the main articles cited and, in some points, mixed sentences make it difficult to understand, as they do not connect.
- “The time of lesions progression…” should be just “Lesion progression…”
Response 4.
Thank you for your feedback. It has been corrected according to your remarks.
Round 2
Reviewer 3 Report
I have the following concern..
Integrins are heterodimers consisting of one alpha and one beta subunit. Therefore, Table 1 should contain not only the name of β2 subunit, but also the name of the α subunit that pairs with β2 subunit.
For example, the beta-integrin subunit CD18 is increased in patients with diabetic retinopathy (DR) [1], and likewise significant increases in alpha-integrin subunits CD11a [2] and CD11b [3] are found in these patients.
Diabetic retinopathy : Mac-1, LFA-1
The above expression is appropriate in table 1.
- Song, L. Wang, and Y. Hui, “Expression of CD18 on the neutrophils of patients with diabeticretinopathy,” Graefe's Archive for Clinical and Experimental Ophthalmology, vol. 245, no. 1, pp. 24–31, 2007.
- Kretowski, J. MyÅ›liwiec, and I. Kinalska, “The alterations of CD11A expression on peripheral blood lymphocytes/monocytes and CD62L expression on peripheral blood lymphocytes in Graves' disease and type 1 diabetes,” Roczniki Akademii Medycznej w Bialymstoku (1995), vol. 44, pp. 151–159, 1999.
- Mastej and R. Adamiec, “Neutrophil surface expression of CD11b and CD62L in diabeticmicroangiopathy,” Acta Diabetologica, vol. 45, no. 3, pp. 183–190, 2008.
Author Response
Response to Reviewer 3 Comments (Round 2)
Thank you for preparing the second review report of our manuscript ID: cells-1213134. All your comments were taken into account.
Point 1.
Integrins are heterodimers consisting of one alpha and one beta subunit. Therefore, Table 1 should contain not only the name of β2 subunit, but also the name of the α subunit that pairs with β2 subunit.
For example, the beta-integrin subunit CD18 is increased in patients with diabetic retinopathy (DR) [1], and likewise significant increases in alpha-integrin subunits CD11a [2] and CD11b [3] are found in these patients.
Diabetic retinopathy : Mac-1, LFA-1
The above expression is appropriate in table 1.
- Song, L. Wang, and Y. Hui, “Expression of CD18 on the neutrophils of patients with diabeticretinopathy,” Graefe's Archive for Clinical and Experimental Ophthalmology, vol. 245, no. 1, pp. 24–31, 2007.
- Kretowski, J. MyÅ›liwiec, and I. Kinalska, “The alterations of CD11A expression on peripheral blood lymphocytes/monocytes and CD62L expression on peripheral blood lymphocytes in Graves' disease and type 1 diabetes,” Roczniki Akademii Medycznej w Bialymstoku (1995), vol. 44, pp. 151–159, 1999.
- Mastej and R. Adamiec, “Neutrophil surface expression of CD11b and CD62L in diabeticmicroangiopathy,” Acta Diabetologica, vol. 45, no. 3, pp. 183–190, 2008.
Response 1:
Thank you for your feedback. We have made the correction to the point.
The β2 integrin subunit (CD18) is able to couple to one of the α subunits (αL-CD11a, αM-CD11b, αX-CD11c, αD-CD11d) to form lymphocyte function-associated antigen-1 (LFA-1/αLβ2/CD11a/CD18), macrophage-1 antigen/complement receptor 3 (Mac1/CR3/ αMβ2/CD11b/CD18), and complement receptor 4 (150,95/CR4//CD11c/CD18).
The beta-integrin subunit CD18 is increased in patients with diabetic retinopathy (DR) [76], and likewise significant increases in alpha-integrin subunits CD11a [77] and CD11b [78] are found in these patients.
- Song, L. Wang, S.L.; Hui, Y. Expression of CD18 on the neutrophils of patients with diabeticretinopathy. Graefe's Arch. Clin. Exp. Ophthalmol. 2007, 245(1), 24–31.
- Kretowski, A.; MyÅ›liwiec, J.; Kinalska, I. The alterations of CD11A expression on peripheral blood lymphocytes/monocytes and CD62L expression on peripheral blood lymphocytes in Graves' disease and type 1 diabetes,” Rocz. Akad. Med. Bialym. 1999, 44, 151–159.
- Mastej, K.; Adamiec, R. Neutrophil surface expression of CD11b and CD62L in diabetic microangiopathy. Acta Diabetol. 2008, 45(3), 183-90.
Studies of lens observed that α3β1, α6β1, α6β4, α1β1, α2β1, α5β1 and αvβ3 integrins are expressed in developing lens [59,60].
The proper names of integrins were corrected in Table 1.
|
Part of the eye/ disease |
Integrins |
|
Cornea/ normal expression injury the healing process of the injury corneal lymphangiogenesis cicatricial pemphigoid epidemic keratoconjunctivitis |
α2β1, α3β1, αvβ5, α6β4, αvβ6 α2, β1 α5β1 α6β4 αvβ1, αvβ5, α3β1 |
|
Conjunctiva/and cornea/ allergic conjunctivitis epidemic keratoconjunctivitis dry eye disease COVID-19 |
α4β1, α4β7, αMβ2, αLβ2 αvβ1, αvβ5, α3β1 αLβ2 αVβ1, αVβ3, αVβ5, αVβ6, αVβ8, α5β1, α8β1, αIIbβ3 |
|
Lens/ normal development cataract |
α3β1, α6β1, α6β4, α1β1, α2β1, α5β1 and αvβ3 αvβ5, αvβ6, αvβ3 |
|
Trabecular meshwork or optic nerve head/ glaucoma |
αvβ3 |
|
Retina/ diabetic retinopathy retinopathy of prematurity regeneration of the axons of the optic nerve sclera/ myopia |
αLβ2 αvβ3 α9β1
α1,β1 |
Table 1. Integrins involved in normal eye and eye diseases
Reviewer 4 Report
The changes in the reviewed version brought significant improvements to the work that seems more interesting to those working in the field.
Author Response
Response to Reviewer 4 Comments (Round 2)
Thank you for preparing the Review Report (Round 2) of our manuscript ID: Cells-1213134. All your comments were taken into account.
Point 1.
The changes in the reviewed version brought significant improvements to the work that seems more interesting to those working in the field.
Answer 1.
Thank you for your feedback and kind words.
English language and style was checked and corrected ny native speaker form our University.